# Effectiveness and Acceptance of Multimodal Antibiotic Stewardship Program: Considering Progressive Implementation and Complementary Strategies

**DOI:** 10.3390/antibiotics9120848

**Published:** 2020-11-27

**Authors:** Flavien Bouchet, Vincent Le Moing, Delphine Dirand, François Cros, Alexi Lienard, Jacques Reynes, Laurent Giraudon, David Morquin

**Affiliations:** 1Service de Maladies Infectieuses et Tropicales, Centre Hospitalier Universitaire de Montpellier, Université de Montpellier, 34000 Montpellier, France; v-le_moing@chu-montpellier.fr (V.L.M.); j-reynes@chu-montpellier.fr (J.R.); d-morquin@chu-montpellier.fr (D.M.); 2Pôle Appui aux Fonctions Cliniques, Département de la Pharmacie, Hôpitaux du Bassin de Thau, 34200 Sète, France; ddirand@ch-bassindethau.fr (D.D.); lgiraudon@ch-bassindethau.fr (L.G.); 3Département Informatique, Hôpitaux du Bassin de Thau, 34200 Sète, France; fcros@ch-bassindethau.fr; 4Département de Biologie Médicale, Hôpitaux du Bassin de Thau, 34200 Sète, France; alienard@ch-bassindethau.fr

**Keywords:** antibiotic stewardship program, complementarity, prospective audit and feedback

## Abstract

Multiple modes of interventions are available when implementing an antibiotic stewardship program (ASP), however, their complementarity has not yet been assessed. In a 938-bed hospital, we sequentially implemented four combined modes of interventions over one year, centralized by one infectious diseases specialist (IDS): (1) on-request infectious diseases specialist consulting service (IDSCS), (2) participation in intensive care unit meetings, (3) IDS intervention triggered by microbiological laboratory meetings, and (4) IDS intervention triggered by pharmacist alert. We assessed the complementarity of the different cumulative actions through quantitative and qualitative analysis of all interventions traced in the electronic medical record. We observed a quantitative and qualitative complementarity between interventions directly correlating to a decrease in antibiotic use. Quantitatively, the number of interventions has doubled after implementation of IDS intervention triggered by pharmacist alert. Qualitatively, these kinds of interventions led mainly to de-escalation or stopping of antibiotic therapy (63%) as opposed to on-request IDSCS (32%). An overall decrease of 14.6% in antibiotic use was observed (*p* = 0.03). Progressive implementation of the different interventions showed a concrete complementarity of these actions. Combined actions in ASPs could lead to a significant decrease in antibiotic use, especially regarding critical antibiotic prescriptions, while being well accepted by prescribers.

## 1. Introduction

In 2015 in Europe, 671,689 cases of infections with antibiotic-resistant bacteria features occurred, leading to 33,110 deaths, corresponding to 6.44 deaths per 100,000 population and 874,541 disability adjusted life-years (DALYs) [1]. Without any practical measures, the current state could worsen exponentially with 390,000 deaths every year expected in Europe by 2050. Moreover, this concerning healthcare issue also represents a dramatic economic burden; that is, if the antibiotic-resistant bacteria infection rate remains at the same level as today, this could lead to a loss of 100 trillion of USD worldwide [2,3]. Implementing antibiotic stewardship programs (ASPs) in hospitals is a major way to improve this issue [4,5,6,7]. Many studies have shown the positive impact of antibiotic stewardship programs (ASPs) on antibiotic use and antibiotic resistance, improvement of morbidity and mortality, reduction of *Clostridium difficile* infections incidence, and health costs savings [7,8,9,10,11,12,13]. One of the key points of ASP success is to gather a multidisciplinary team including pharmacists, microbiologists, and infectious diseases physicians with a specific time dedicated to this task [5,14,15].

Nowadays, cross-disciplinary medical project funding is limited by the current economic healthcare situation. Despite many warnings from French and European infectious diseases societies about the crucial need for ASPs, raising funds and dedicating time to implement these strategies are still difficult, especially when the short-term economic benefit is not obvious [16,17,18].

In France, ASPs are not fully implemented and the current system relies on supporting prescribers mainly through training and on-request infectious diseases specialist consulting [19]. Training may be a key point to improve antibiotics prescription, yet a recent multicenter web-based survey brings to light that most final-year European medical students feel they still need more education on antibiotic use for their future practice as junior doctors [20]. In this context, the association of the improvement of medical student training and a more interventionist strategy including microbiological laboratory alerts and prospective audit and feedback (PAF) interventions, such as prescription review with assistance by pharmacists, could be useful [6,21,22]. Indeed, PAF allows clinicians to prescribe any empiric antibiotic regimen, then the ASP can advise the clinician on discontinuing or adjusting therapy after prescription analysis. Although feedback further increased the intervention effect, it is used in only a minority of enabling interventions, as shown in a Cochrane meta-analysis [23]. This study raises the need for new studies to assess different stewardship interventions and to explore the facilitators to implementation.

Indeed, the practical way to link together these interventions is not clear and neither the complementarity of these actions nor the acceptance of physicians towards unsolicited advice have yet been evaluated. Based on recent publications [24,25,26,27], we progressively implemented an innovative multimodal ASP in 2018 in a secondary care hospital. The aim of this study was to evaluate the complementarity of different interventions in an ASP and the impact on antibiotic use. We also analyzed the impact on mean of length of stay (LOS), 30-day readmission rate (30-DRR), and mortality. Prescribers’ acceptance was also assessed in the perspective of long-term system development.

## 2. Results

### 2.1. Interventions Complementarity

Over the entire analysis period, 7508 stays involved the administration of antibiotic therapy, of which 1316 received an intervention. At least one intervention was carried out for 1430 stays, corresponding to 2046 interventions noted in the electronic medical records (EMRs) (Figure 1).

The overall acceptance rate for the proposals was 88%, with a variation according to intervention types ranging from 68 to 92% (Figure 2). The distribution analysis of intervention-types normalized to working days on site highlights a true complementarity between interventions (Figure 3). In summary, the implementation of PAF interventions in a second phase widens the ASP field of action without impacting other types of intervention.

This complementarity is also illustrated when comparing advice type according to different kinds of interventions. Regarding IDSCS interventions, 341 of 1053 interventions (32%) were either a lack of antibiotic initiation, a therapeutic de-escalation, a cessation of all antibiotics, or a reduction in treatment duration with a 92% acceptance rate. On the other hand, 535 proposals (51%) were either a therapeutic escalation or an extension of antibiotic therapy with a 92% acceptance rate.

Of the 501 proposals made during PHARM-cATB interventions, 316 (63%) were either a therapeutic de-escalation, a cessation of all antibiotics, or a reduction in the duration of treatment with a 76% acceptance rate. Only 69 proposals (14%) consisted of a therapeutic escalation or an extension of antibiotic therapy with an 84% acceptance rate (Figure 4).

### 2.2. Impact on Mortality, 30-Day Readmission Rate, and Mean Length of Stay

There were 3561 inpatients with deep infections hospitalized in the eight wards who benefited from the whole ASP from January 2016 to May 2017 versus 3839 from January 2018 to May 2019. The clinical and demographic characteristics of these patients are summarized in Table 1. There was a downward trend in the mean LOS in patients with deep infections, from an LOS of 11.03 days before the implementation of the system to a LOS of 10.44 days after implementation, but this difference was not statistically significant (*p* = 0.096). Nor was there any significant difference regarding in-hospital mortality in patients with deep infections (267 (7.31%) versus 266 (6.9%); *p* = 0.37) or 30-DRR (8.2% (293) versus 7.7% (296) in 2019, *p* = 0.44).

### 2.3. Impact on Antibiotic Consumption

Overall, antibiotic use was significantly decreased by 14.6% in the whole hospital after ASP implementation (336 daily dose of antibiotics per 1000 patient-days (DDD_1000PD_) in 2017 versus 287 DDD_1000PD_ in 2019; *p* = 0.03). Carbapenems use was moderate and stable over time (from 5 DDD_1000PD_ in 2017 to 4 DDD_1000PD_ in 2019; *p* = 0.82). A slight increase in injectable third-generation cephalosporins use was observed (from 53 DDD_1000PD_ in 2017 to 60 DDD_1000PD_ in 2019; *p* = 0.12). There was a significant decrease of fluoroquinolones use of 63% (51 DDD_1000PD_ in 2017 versus 19 DDD_1000PD_ in 2019; *p* = 0.03) (Figure 5). We also observed a significant decrease in overall antibiotic use for the eight departments included in PAF interventions from 543 DDD_1000PD_ in 2017 versus 474 DDD_1000PD_ in 2019 (*p* = 0.016). Moreover, the reduction in fluoroquinolones use was more noticeable between April and December 2018 (60 DDD_1000PD_ versus 34 DDD_1000PD_, respectively). This decrease continued until the end of the analysis. An effect on carbapenems use took longer to appear, but a clear decrease was observed from January to June 2019 (13 DDD_1000PD_ versus 7 DDD_1000PD_, respectively) (Figure 6).

### 2.4. User Experience Assessment: Satisfaction Survey

Ninety-five physicians that participated in the ASP were surveyed for satisfaction, of which 49 responded. All physicians were satisfied with the dedicated phone line provided and wanted on-request IDSCS to be continued, as well as LAB-M interventions. Regarding PAF actions, only 2/32 physicians were not satisfied with this kind of intervention and did not wish for it to be carried on. All results are summarized in Appendix A.

## 3. Discussion

We demonstrated that implementation of a whole ASP combining solicited and unsolicited interventions is possible and that the different modes of intervention are complementary. The multimodal ASP implemented in the hospital of Bassin de Thau (HBT) led to a decrease of antibiotic use, especially fluoroquinolones, without impacting deep infection mortality. We also observed a slightly decreased trend in the length of hospital stay in these patients.

Many ASP strategies have already shown their efficacy; for example, development and implementation of facility-specific clinical practice guidelines for common infectious diseases syndromes, IDS systematic referral for *Staphylococcus aureus* bacteremia clinical cases, PAF, preprescription authorization for certain antibiotics, specific interventions depending on infection type or clinical department, microbiological laboratory interventions, and so on [10,11,24,28,29]. To the best of our knowledge, our study is the first to evaluate the potential synergy between different kinds of actions. Our main contribution is to demonstrate that multimodal interventions are synergic. Indeed, opposite to former studies evaluating the impact of specific different antibiotic stewardship interventions, our global approach highlighted the complementarity of each intervention in the success of the holistic ASP. It is noteworthy that this study was set up in the French healthcare system where interfering methods are not developed [19]. Thanks to the progressive implementation of this system, we were able to highlight the complementarity of the interventions. Indeed, the on-request IDSCS, LAB-M, and ICU-M interventions number was stable overtime, even after implementation of PAF strategies, i.e., PHARM-cATB and PHARM-7d review. This highlights a cumulative effect between the different modes of intervention, suggesting that each kind of intervention responded to a specific type of problem. Indeed, on-request IDSCS led to escalation or lengthening of antibiotic therapy in more than 50% of cases, while PHARM-cATB resulted in escalation or lengthening in only 14%. Conversely, PHARM-cATB reviews resulted in de-escalation, stopping, or shortening of antibiotic therapy in 60% of cases. This proportion was even higher within PHARM-7d review. Thus, the different modes of intervention were complementary, both qualitatively and quantitatively.

Antibiotic use analysis also revealed the same pattern. Interestingly, the decreased consumption accelerated after implementation of PAF methods, suggesting a stronger impact of interfering methods on overall antibiotic use. Similar results were found by Tamma et al., whose study highlighted the effectiveness of this method and its major impact on antibiotic use in a cross-over trial [30]. According to these results, a recent retrospective study analyzed the impact of interventionist strategies as PAF or preprescription authorization on fluoroquinolone consumption in 48 U.S. hospitals [31]. Fluoroquinolone use was significantly decreased by 26% over two years between establishment with ASP targeting fluoroquinolone and those with no ASP.

We did not find evidence of any statistically significant differences on mean LOS, 30-DRR, or mortality between the two periods. However, we observed a downward trend in the mean LOS with a decrease of 0.6 days of hospitalization per stay. Indeed, the reduction in antibiotic consumption, particularly intravenous antibiotics, might lead to a reduction in adverse effects and an earlier discharge of patients. The absence of statistical significance might be because of a lack of power for this criteria; however, it was not the primary endpoint of this study. In the literature, some arguments tend to confirm this hypothesis: Sasikumar et al. showed a significant impact of IDS interventions on mortality and medical stay costs, especially for ICU stays [32]. Although there was no significant positive impact on mortality, 30-DRR, and mean LOS in our study, we did not observe any negative impact on patients’ clinical outcomes. Moreover, the high prevalence of patients with chronic respiratory failure, end-stages renal diseases, immunosuppression, and diabetes in the second analysis period could lead to an underestimation of the potential positive impact of our ASP on these outcomes.

These data highlight the importance of using a multimodal strategy when setting up an ASP, keeping in mind that different interventions would respond to different needs. The 2016 IDSA guidelines emphasized PAF and preprescription authorization methods, while underlining the potential for better acceptance of PAF as prescriber autonomy is maintained [24]. In our study, the acceptance rate of PAF intervention was high (79%), despite the fact that unsolicited specialist consulting is not culturally ingrained into the French medical community. Most physicians interviewed in the satisfaction survey agreed that on-request IDSCS and LAB-M actions were improving clinical outcomes and should be continued, whereas only two physicians viewed unsolicited interventions as intrusive to their practice and were reluctant to maintain these methods. PAF acceptance was probably better than expected thanks to its implementation over a second phase of the program, whereas more conventional methods, i.e., on-request IDSCS and LAB-M, were already set up. So, sequential implementation can be identified as a facilitator regarding acceptance of interventionist methods. These results reinforce IDSA recommendations to develop and promote PAF strategies.

We show that they may be implemented within French hospitals considering their efficiency and their complementarity to other methods. It is important to note that, without an EMR, it is challenging to set up such a program with unsolicited interventions.

Nevertheless, this kind of program is time-consuming and labor-intensive; indeed, PAF interventions represented 10 h of work per week for one IDS and one pharmacist, while LAB-M interventions counted for 5 h of work, without including intervention retranscription in the electronic patient record (about 30 min for each intervention, i.e., 25 h weekly). This organization requires dedicated medical time for this activity, as recommended by European Society of Clinical Microbiology and Infectious Diseases (ESCMID) [26].

There are limitations to our study. The impact on antibiotic resistance was not assessed because of the short-term study design. This key outcome will be analyzed after several years of operating under the program in order to compare antimicrobial resistance before and after implementation of this system. Our study did not include medico-economic analysis. Nevertheless, the 0.6 days of stay decrease for inpatients with ID diagnosis would allow some healthcare cost saving, despite this result not being statistically significant. We were also not able to set up, in parallel to our ASP, an educational program that could lead to improved practitioner adherence as well as antibiotics prescribing over the long term [33]. Indeed, in a recent Spanish study, the quality of antimicrobial prescribing improved markedly, and the inappropriate treatment rate was significantly lower over 3 years thanks to regular educational interviews [34]. In addition, we could not evaluate antibiotic prescription at the discharge because of the lack of computerization. Indeed, Vaughn et al. recently highlight that hospital-based stewardship interventions did not affect antibiotic prescription at the discharge [31]. In this study, 14/48 hospitals reported using pre-prescription approval and/or PAF to target fluoroquinolone prescriptions, but hospitals with fluoroquinolone stewardship had twice as many new fluoroquinolone starts after discharge as hospitals without. Weber et al. analyzed discharge prescriptions in a 576-bed academic hospital in Portland. Among 6701 discharges, 22.9% were prescribed antibiotics upon discharge [35]. To complete these data, Scarpato et al. analyzed the appropriateness of antimicrobial agents prescribed on discharge [36]. They found that 70% of discharge antibiotics were inappropriate in antibiotic drug choice, dose, or duration. Analysis of discharge prescriptions should be the next step of our ASP with the implementation of an educational program to improve the prescription of discharge antibiotics. Moreover, there are biases inherent to the design of our study. Indeed, as for many “before–after” studies, the two groups we compared are heterogeneous. However, we found more pre-existing medical conditions for patients in the period after implementation of our ASP; therefore, this might lead to underestimation of the impact on the mean LOS downward trend we observed.

Finally, there was a center’s effect limiting the extrapolation of our results as our study took place in a small hospital with less than 300 beds for the medicine, surgery, and obstetrics departments. The small hospital size likely facilitated the rapid establishment of this multidisciplinary system. One of the reasons of our success is probably the direct and confident relationship established between the IDS, the pharmacist, the microbiologist, and the prescribers, which may not be possible to install in other settings. Additional multicentric studies are needed to confirm our results and go further.

## 4. Materials and Methods

### 4.1. Study Setting and Interventions

Two hospitals (secondary care hospital of Bassin de Thau (HBT) and university hospital of Montpellier (UHM)) created a shared infectious disease specialist (IDS) position to sequentially implement an innovative multimodal ASP within HBT. The HBT is a 938-bed hospital with establishments providing care for dependent elderly people (376 beds); psychiatric unit (57 beds); geriatric and follow-up care and rehabilitation unit (167 beds); and acute care unit, medicine, surgery, and obstetrics unit (274 beds). On the whole, 406 beds are provided with an electronic medical record (EMR).

The infectious disease EMR pattern was duplicated from UHM to HBT to allow IDS response in real time with a complete traceability in the patient EMR for each intervention [37].

Several interventions centralized by the same IDS were progressively implemented:

January 2018: Simultaneous implementation of (i) a dedicated phone line for the infectious diseases specialist consulting service (IDSCS), (ii) weekly intensive care unit multidisciplinary clinical team meetings (ICU-M), and (iii) IDS intervention triggered by a bi-weekly microbiological laboratory meeting (LAB-M) for the revision of antibiotics based on microbiological data (blood cultures, per-operative samples, lumbar, pleural, and joint punctures). In addition, monthly educational training on antibiotic use was proposed to all residents of the hospital.

April to December 2018: Establishment of PAF interventions in association with pharmacy unit members. Initiation of critical antibiotics prescription review (PHARM-cATB) in April 2018. This consists of a systematized analysis of critical antibiotics prescription (injectable third-generation cephalosporins, fluoroquinolones (FQ), and carbapenems) performed twice a week within the eight wards with the greatest antibiotics use, with feedback to the prescriber.

Initiation of longer than 7 days antibiotics prescription review (PHARM-7d) in December 2018. A systematized analysis was performed on the whole hospital with feedback to the prescriber.

Each intervention was noted in the EMR in real time and was analyzed to evaluate intervention acceptance. An intervention was defined as having been followed if the proposed antibiotic type, duration, and dosage were accepted by the prescribing physician. The whole system organization is summarized in Figure 7.

### 4.2. Outcomes

The complementarity of the different actions was assessed by the quantitative and qualitative analysis of all interventions traced in the EMR (number of different types of interventions over time, analysis of proposal for each intervention, and impact of interventions on antibiotic use). Antibiotic consumption, defined in daily dose of antibiotics per 1000 patient-days (DDD_1000PD_), was calculated with ConsoRes^®^ software [38] for the years 2016–2019 in the whole hospital. The impact on mean LOS, 30-DRR, and in-hospital mortality was assessed on patients with deep infections from the eight wards (medicine, surgery, and intensive care unit) representing 274 acute care beds that benefited from all the interventions of ASP by comparing two groups of patients over two periods of 18 months: January 2016–May 2017 versus January 2018–May 2019. All patients who were diagnosed with deep infections based on the International Classification of Diseases, Information System Medicalization Program were included in this comparison.

### 4.3. User Experience Assessment

An anonymous satisfaction survey was sent after 12 months of implementation of the system to all clinicians, followed by two reminder letters.

### 4.4. Statistical Approach

Comparisons between the two periods were made using a χ^2^ test for qualitative variables. Comparisons between the two periods were performed using a χ^2^ test for mortality and readmission rate, a Mann–Whitney U test for the mean LOS, and a linear regression test to analyze antibiotic consumption.

### 4.5. Ethics

This study was conducted according to the principles of the declaration of Helsinki and in compliance with International Conference on Harmonization/Good Clinical Practice regulations. According to the French law, the study was in accordance with the recommendations of the local ethics committee, without the need for consent.

## 5. Conclusions

This study is among the first to analyze the complementarity and impact of combining different strategies, especially interventionist methods, developed within ASPs. This system set up with reasonable human resources could easily be transposable to size-equivalent hospitals. A good acceptance rate of PAF interventions and clear complementarity of the different types of actions, leading to a major decrease in fluoroquinolones use and overall antibiotic use, without a negative impact on mortality or 30-DRR, are key points of this study.

Further studies are needed to strengthen the scope of our results, including multidisciplinary and educational programs; long-stay healthcare structures; analysis of discharge prescriptions; and giving a more important role to PAF interventions, which currently are likely not sufficiently developed [39,40]. As this type of system is probably cost-effective, the economic aspect should not be an obstacle to its implementation.

## Figures and Tables

**Figure 1 antibiotics-09-00848-f001:**
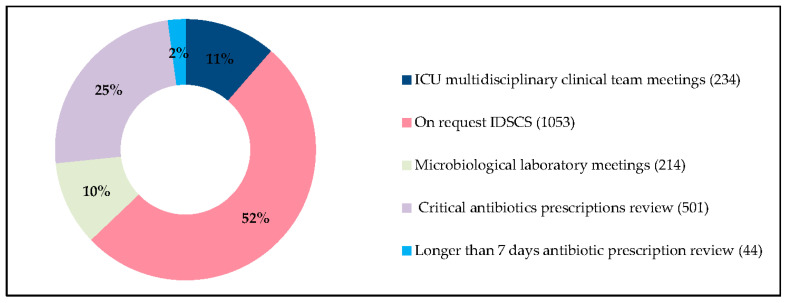
Interventions’ distribution. Distribution of the 2046 interventions noted in the electronic patient record related to 1430 hospital stays and 1243 patients. Abbreviations: ICU, intensive care unit; IDSCS, infectious diseases specialist consulting service.

**Figure 2 antibiotics-09-00848-f002:**
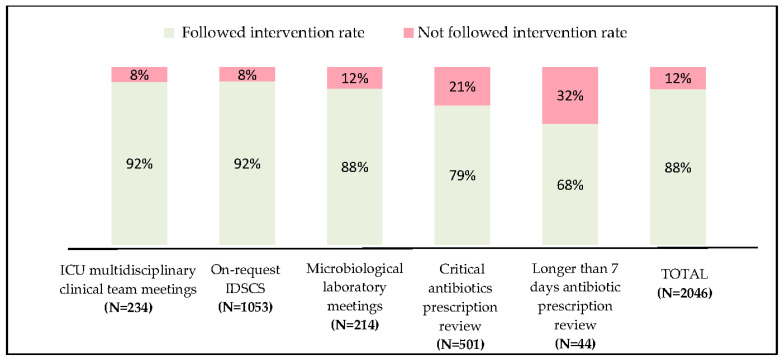
Compliance depending on intervention type. Compliance rate of the 2046 interventions noted in electronic patient record. Abbreviations: ICU, intensive care unit; IDSCS, infectious diseases specialist consulting service; N = number of interventions.

**Figure 3 antibiotics-09-00848-f003:**
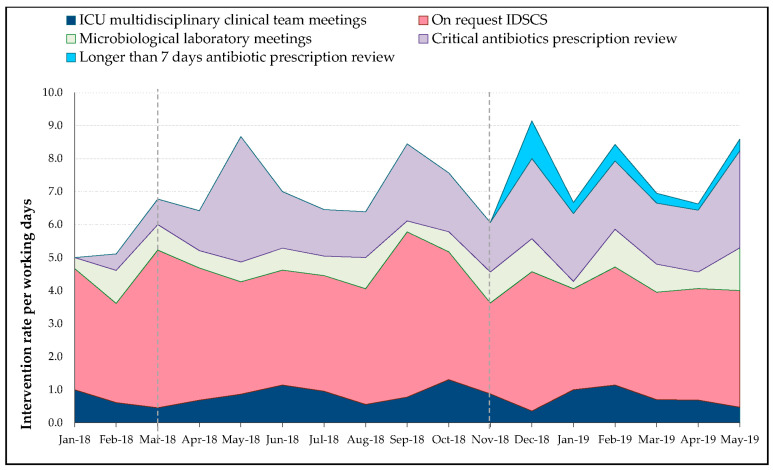
Interventions’ distribution per working days. Dotted lines represent initiation of prospective audit and feedback interventions in association with pharmacy unit members (critical antibiotics prescription review and longer than 7 days antibiotic prescription review, respectively). Abbreviations: ICU, intensive care unit; IDSCS, infectious diseases specialist consulting service.

**Figure 4 antibiotics-09-00848-f004:**
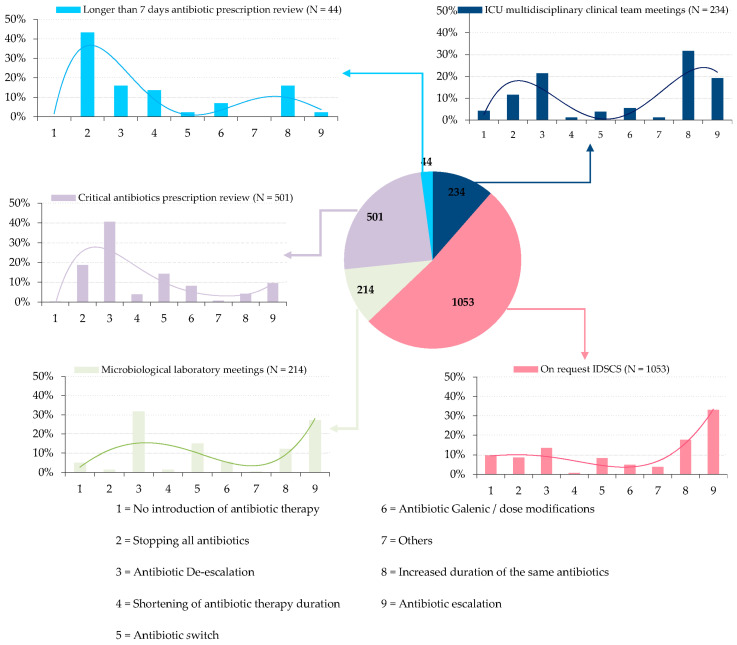
Advice type according to different kind of interventions. Distribution of the 2046 interventions noted in the electronic patient record related to 1430 hospital stays and rate of propositions according to different kind of interventions. N = number of interventions. IDSCS = infectious diseases specialist consulting service.

**Figure 5 antibiotics-09-00848-f005:**
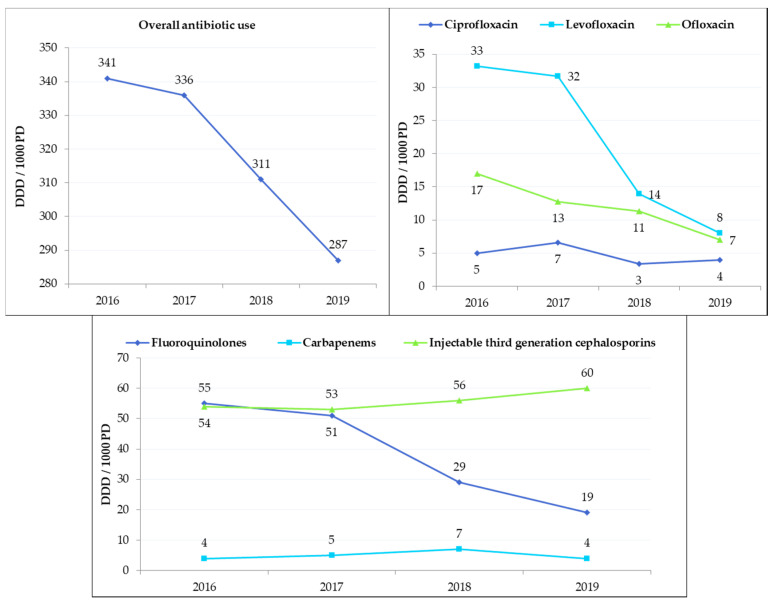
Evolution over time of antibiotic consumption in secondary care hospital of Bassin de Thau. Data are presented as defined daily dose of antibiotics per 1000 patient-days (DDD_1000PD_).

**Figure 6 antibiotics-09-00848-f006:**
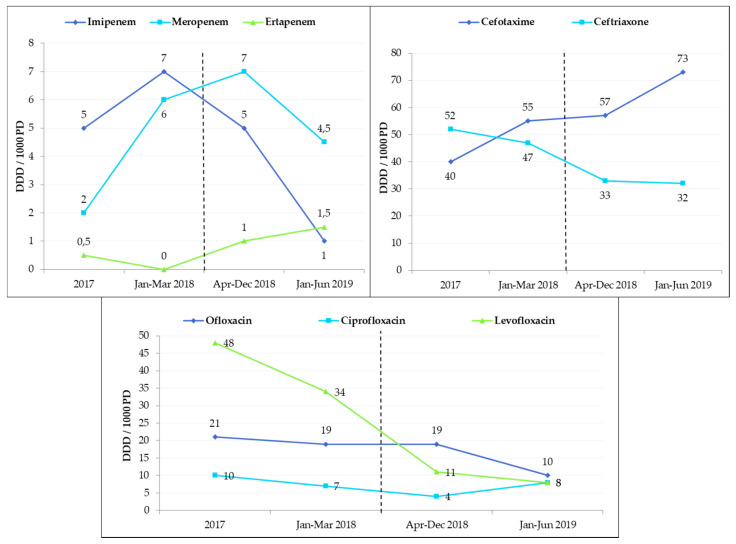
Evolution over time of antibiotic consumption in the eight units that tested the prospective audit and feedback intervention (antibiotics prescription review). Dotted lines represent initiation of critical antibiotics review in association with pharmacy unit members. Data are presented as defined daily dose of antibiotics per 1000 patient-days.

**Figure 7 antibiotics-09-00848-f007:**
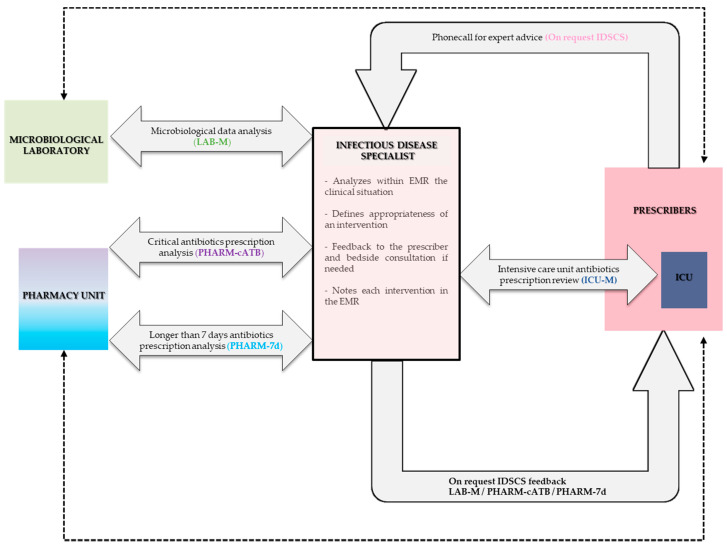
Antibiotic stewardship program organization in the studied hospital when fully established in December 2018. Dotted lines represent pre-existent relationship before implementation of the antibiotic stewardship program. On request IDSCS: on request infectious diseases specialist consulting service via a dedicated phone line. ICU-M: intensive care unit multidisciplinary clinical team meetings for weekly antibiotics prescription review in the ward. LAB-M: microbiological laboratory meetings for bi-weekly analysis of microbiological samples with microbiologists and infectious diseases specialist. PHARM-cATB: critical antibiotics prescription review twice a week with pharmacist and infectious diseases specialist. PHARM-7d: longer than 7 days antibiotic prescriptions review twice a week with pharmacist and infectious diseases specialist. EMR: electronic medical records. ICU: intensive care unit.

**Table 1 antibiotics-09-00848-t001:** Demographic characteristics and pre-existing medical conditions of patients with a diagnosis of deep infection before and after implementation of the antibiotic stewardship program in the eight wards that benefited from all types of interventions.

	Before Implementation	After Implementation	*p*-Value
	1/1/2016 to 31/5/2017	1/1/2018 to 31/5/2019	
**Number of stays**	3561	3839	
**Gender**			
Female	1646 (46%)	1848 (48%)	
Male	1915 (54%)	1991 (52%)	
**Age (years)**			
Mean (Min–Max)	73.13 (17–108)	73.30 (17–103)	
**ICU stays**			
	600 (17%)	567 (15%)	0.015
**Pre-existing medical conditions**			
Solid organ transplant	7 (0.2%)	14 (0.4%)	0.25
Immunomodulatory therapy	1 (0.03%)	9 (0.2%)	0.036
End stages renal disease (IV–V)	30 (0.8%)	72 (1.9%)	0.0002
Chronic liver disease	47 (1.3%)	39 (1%)	0.27
Chronic respiratory failure	187 (5.3%)	250 (6.5%)	0.025
Agranulocytosis	10 (0.3%)	15 (0.4%)	0.54
Chemotherapy during the stay	2 (0.06%)	9 (0.2%)	0.09
Diabetes	724 (20%)	874 (23%)	0.01
HIV	33 (0.9%)	24 (0.6%)	0.18
**Infection types**			
Pyelonephritis	1124	1013	
Intra-abdominal infections	651	673	
Cellulitis and skin abscess	233	253	
Meningitis	5	7	
Endocarditis	9	10	
Pulmonary infection	1834	1809	
Osteomyelitis and prosthetic joint infection	62	70	

Results are presented as No (and rate %). All patients with a diagnosis of deep infection regarding the International Classification of Diseases were included. Abbreviation: HIV, human immunodeficiency virus; ICU, intensive care unit.

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
