# Peer review of "Effectiveness and Acceptance of Multimodal Antibiotic Stewardship Program: Considering Progressive Implementation and Complementary Strategies"

_antibiotics, 2020, doi:10.3390/antibiotics9120848_

Round 1
Reviewer 1 Report
This study assesses complementarity of the different cumulative actions through quantitative and qualitative analysis of all interventions traced in the electronic medical record. They observed a quantitative and qualitative complementarity between interventions directly correlate to a decrease in antibiotic use. My recommendation for this paper is to publish after major revision.
Some comments which greatly enhance the understanding of the paper and its value are presented below. Specific issues that require further consideration are:
- In a paragraph starting in L208 on P10, the authors claims that ”Absence of statistical significance might be due to a lack of power of this study”. It is regrettable that they are cherry-picking results of statistical tests. The poor statistical power also raises serious concerns about false negative rates on the testing procedures they employed, and undermines the reliability of their results.
- Their study design has no consideration of confounding effects. For example, in Table 1 they provided numerical summary of two groups (before and after). The two groups are heterogeneous in many ways: years, conditions of patients, etc. It is questionalble that any statistical difference between groups are caused by the implementation or some hidden unoberved factor.
- The current state of knowledge relating to the manuscript topic has been presented, but the author’s contribution and novelty are not enough emphasized.
Author Response
We would like to take this opportunity to thank the referee for all her/his comments and help in improving the quality of this article.
-Comment 1:
In a paragraph starting in L208 on P10, the authors claims that ”Absence of statistical significance might be due to a lack of power of this study”. It is regrettable that they are cherry-picking results of statistical tests. The poor statistical power also raises serious concerns about false negative rates on the testing procedures they employed, and undermines the reliability of their results.
We agree that this sentence can be confusing. However, our study was not designed to show statistical differences between the two groups for mean of length of stay (LOS), 30-days readmission rate (30-DRR) or mortality because there were not primary endpoints. Moreover, regarding in-hospital mortality and 30-DRR in patients with deep infections, we found similar results: 267 deaths versus 266 and 293 readmissions versus 296. Thus, we can reasonably assume the testing procedures do not undermine the reliability of these results.
This sentence has been modified as follow in the manuscript in the Discussion part L216 on P10:
“Absence of statistical significance might be due to a lack of power for this criteria however it was not the primary endpoint of this study”.
-Comment 2:
Their study design has no consideration of confounding effects. For example, in Table 1 they provided numerical summary of two groups (before and after). The two groups are heterogeneous in many ways: years, conditions of patients, etc. It is questionalble that any statistical difference between groups are caused by the implementation or some hidden unoberved factor.
We agree that there are differences between patients in the two periods but they are modest and tend to be more severe in the second period (more diabetes, respiratory failure, immunomodulatory therapy and chemotherapy during the stay). Therefore, these differences are not likely to have influenced the trend towards shorter length of stay in the second period. For this reason, a multivariate analysis did not seem relevant to us, especially since this is not the main result of the work.
This sentence has been added in the manuscript in the Discussion part L266 on P11:
“Moreover, there are bias inherent to the design of our study. Indeed, as for many “before-after” studies, the two groups we compared are heterogeneous. However, we found more preexisting medical conditions for patient in the period after implementation of our ASP, thus, this might lead to underestimate the impact on the mean LOS downward trend we observed.”
-Comment 3:
The current state of knowledge relating to the manuscript topic has been presented, but the author’s contribution and novelty are not enough emphasized.
According to the reviewer comment, these sentences have been added/modified in the manuscript as follow:
In the Introduction part L76 P2:
“The aim of this study was to evaluate the complementarity of different interventions in an ASP and the impact on antibiotic use. We also analyzed the impact on mean of length of stay (LOS), 30-days readmission rate (30-DRR) and mortality. Prescribers’ acceptance was also assessed in the perspective of long-term system development.”
In the Discussion part L190 P10
“Our main contribution is to demonstrate that multimodal interventions are synergic. Indeed, opposite to former studies evaluating impact of specific different antibiotic stewardship interventions, our global approach highlighted the complementarity of each intervention in the success of the holistic ASP”
Reviewer 2 Report
This is an interesting study concerning multimodal antibiotic stewardship program (ASP) comparing two time intervals: Jan 2016 to May 2017 versus Jan 2018 to May 2019. The authors compared the results before and after implementation of the multimodal ASP.
The authors described well the different intervention types of ASP and their compliance.could found an significant, overall decrease in antibiotic use, which was carefully specified in the specific antibiotic classes.
This multimodal approach also could demonstrate that the different modes of intervention were complementary and synergistic.
The study is well designed and presented.
Author Response
We would like to take this opportunity to thank the referee for all her/his comments and help in improving the quality of this article.
Reviewer 3 Report
The authors analyzed the complementarity and impact of combining different strategies, especially interventionist methods developed in antibiotic stewardship program. This study showed good acceptance rate of prospective audit and feedback interventions, clear complementarity of the different types of actions, leading to major decrease in fluoroquinolones use and overall antibiotic use, without a negative impact on mortality or 30-days readmission rate. The authors also acknowledged the limitations and further studies needed.
Overall, it is well written. Noticed the sentence in the manuscript template was not deleted in lines 341-342.
Author Response
We would like to take this opportunity to thank the referee for all her/his comments and help in improving the quality of this article.
Comment 1:
“Noticed the sentence in the manuscript template was not deleted in lines 341-342.”
We removed the sentence L356-357:
This section is not mandatory, but can be added to the manuscript if the discussion is unusually long or complex.
Reviewer 4 Report
The manuscript "Effectiveness and Acceptance of Multimodal Antibiotic Stewardship Program: Considering Progressive Implementation and Complementary Strategies" by Bouchet et al. describes the multimodal antibiotic stewardship with the implementation of four combined modes of interventions. The paper is well written and the results are clearly showed, but there are some important considerations that have to be taken into account.
The section 2.4 treats the satisfaction survey, and this represents an objectively useful data if supported by a high number of participants. The authors confirmed that only 32 of 154 physicians responded. The reliability of the data is not very high due to the low number of responses, so it advisable to remove these results or to support them with a greater number of the responses.
In the discussion section (line 250-251) the authors confirmed that they did not evaluate antibiotic prescription at the discharge. Even if this the basis and the development of this type of system look very prominent, it is necessary to consider the therapies after discharge to better evaluate the cost-benefit ratio.
In the conclusion section (line 341-342) remove the sentence.
Round 2
Reviewer 1 Report
I am OK with their responses. My recommendation for the paper is to be published in present form.
Reviewer 4 Report
The manuscript can be accepted in the present form for the publication.